# PhysForge: Generating Physics-Grounded 3D Assets for Interactive Virtual World

**Yunhan Yang** [* 1 2]  **Chunshi Wang** [* 3 2]  **Junliang Ye** [* 4 2]  **Yang Li** [2]  **Zanxin Chen** [5]  **Zehuan Huang** [6]  **Yao Mu** [5]  **Zhuo Chen** [2]  **Chunchao Guo** [2]  **Xihui Liu** [1]

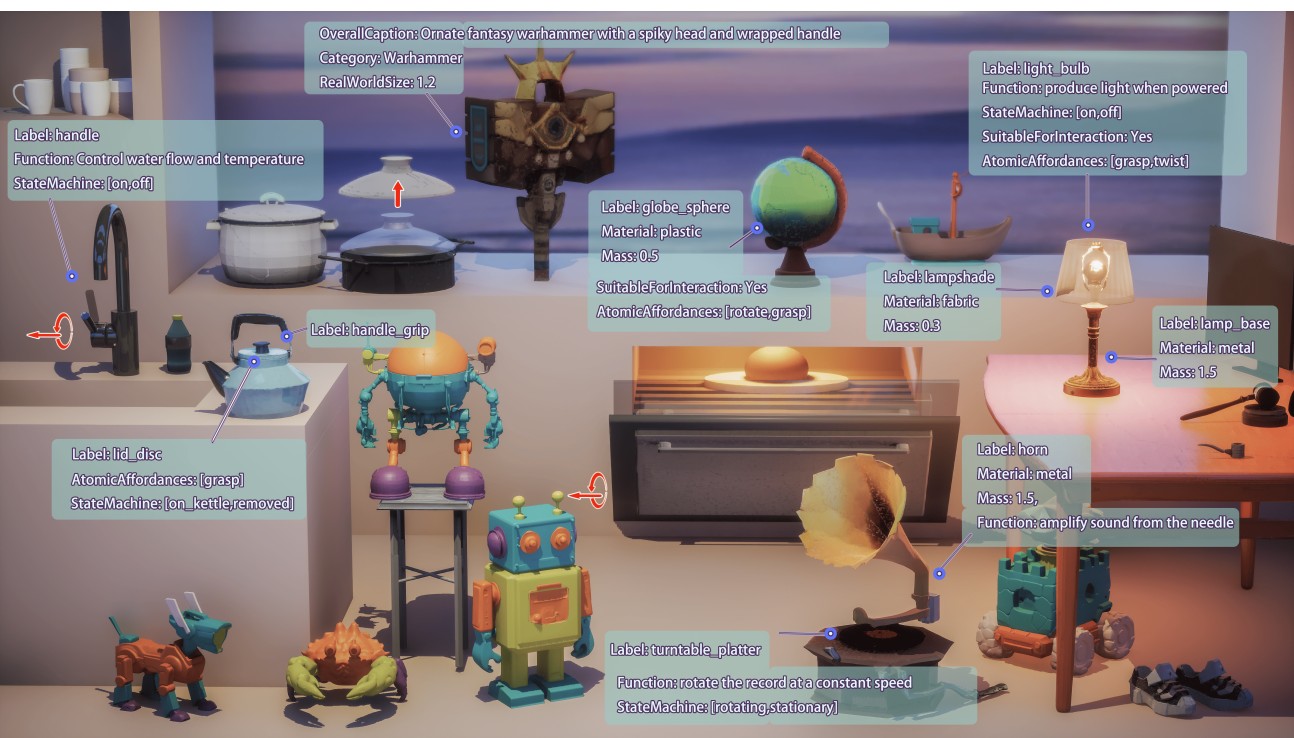

*Figure 1.* PhysForge takes only a single input image to generate physics-grounded 3D assets. The figure showcases our high-quality generated results, where: (a) parts are distinguished by different colors to show their geometry; (b) parts are annotated with their detailed physical properties (text labels); and (c) the kinematic parameters for movable parts (such as joint axes) are precisely indicated by arrows. Our assets are ready for building interactive virtual worlds.

## Abstract

Synthesizing physics-grounded 3D assets is a critical bottleneck for interactive virtual worlds and embodied AI. Existing methods predominantly focus on static geometry, overlooking the functional properties essential for interaction. We propose that interactive asset generation must be rooted in functional logic and hierarchical physics.

To bridge this gap, we introduce PhysForge, a decoupled two-stage framework supported by PhysDB, a large-scale dataset of 150,000 assets with four-tier physical annotations. First, a VLM acts as a "physical architect" to plan a *"Hierarchical Physical Blueprint"* defining material, functional, and kinematic constraints. Second, a physics-grounded diffusion model realizes this blueprint by synthesizing high-fidelity geometry alongside precise kinematic parameters via a novel *KineVoxel Injection* (KVI) mechanism. Experiments demonstrate that PhysForge produces functionally plausible, simulation-ready assets, providing a robust data engine for interactive 3D content and embodied agents.

*Equal contribution [1]The University of Hong Kong [2]Tencent Hunyuan [3]Zhejiang University [4]Tsinghua University [5]Shanghai Jiao Tong University [6]Beihang University. Correspondence to: Xihui Liu <xihuiliu@eee.hku.hk>, Chunchao Guo <chunchaoguo@gmail.com>.

*Proceedings of the 43 $^{rd}$ International Conference on Machine Learning*, Seoul, South Korea. PMLR 306, 2026. Copyright 2026 by the author(s).

# 1. Introduction

Recently, 3D generative models have achieved rapid progress, capable of synthesizing 3D assets with diverse appearances and high-fidelity geometric details (Zhang et al., 2023; Xiang et al., 2024). Concurrently, embodied AI and virtual game environments face a soaring demand for large-scale, high-quality 3D content. 3D generation technology holds the promise to serve as a *data engine* to this content bottleneck. However, a significant gap remains: the vast majority of existing 3D generation methods focus solely on generating *static* geometry and textures, overlooking the physics information that is crucial for interaction. These generated "hollow shell" assets cannot be grasped, pushed, or manipulated by agents, making them difficult to deploy directly in embodied AI simulators or game environments that require realistic physical interactions. To bridge this gap, we aim to propose a generation pipeline capable of producing *physics-grounded* 3D assets directly. Our core insight is that for an object to be physically interactive, its generation must be driven by its *functional logic* and *hierarchical physics*. For example, a button on a television is the basic unit of function and operation; a cabinet's door and handle each carry distinct materials, functions, and kinematic definitions. Therefore, we shift the focus from traditional holistic shape generation to physics-centric synthesis, where the object's structure is a manifestation of its intended physical functions.

To achieve this, we propose **PhysForge**, an innovative two-stage framework that decouples physical planning from physical realization. Inspired by the "planning-then-generation" paradigms successful in 2D multimodal research (Sun et al., 2024; Chen et al., 2025a), our design leverages the complementary strengths of specialized generative architectures: while VLMs possess the world knowledge necessary for complex physical planning, diffusion models excel at the precise synthesis of kinematic parameters, geometry, and textures. By decoupling these processes, PhysForge ensures that the generated assets are not only visually realistic but also physically consistent and simulation-ready.

The first stage is **VLM-based Planning**. Instead of starting from scratch, we finetune a powerful VLM, enabling it to acquire 3D spatial understanding and part-structure planning capabilities while retaining its inherent world knowledge. This VLM takes an image, an optional 2D mask, and generated 3D voxels (Xiang et al., 2024) as input, and is tasked with generating what we call *"Hierarchical Physical Blueprints"*. This blueprint includes the bounding box layout for all parts, as well as detailed physical properties for each part (including parent nodes, articulation types, etc.). We discover a critical synergistic effect: the introduction of physical properties, in turn, significantly aids the model's structural planning. By providing functional and physical constraints, it effectively resolves the ambiguity of part granularity, allowing the model to produce reasonable part decompositions even without 2D mask guidance.

The second stage is **Diffusion-based Generation**. After obtaining the *blueprint*, we meticulously "forge" the high-fidelity geometry alongside the precise kinematic parameters promised in the planning stage. We propose a novel KineVoxel Injection (KVI) mechanism. This method encodes precise articulation parameters (like origin, axis, and limit) into a special *kinematic voxel*, allowing it to be jointly generated with the geometry-representing voxels during the diffusion denoising process, thereby achieving a synergistic synthesis of geometry and kinematic parameters.

To train our model effectively, we construct and introduce PhysDB, a large-scale dataset containing 150k assets. We define a novel four-tier annotation system that captures physics hierarchically. The holistic tier defines global properties like real-world scale and usage scene (e.g., kitchen, bedroom). The static properties tier covers part-level attributes such as semantic labels, physical materials (e.g., "metal", "wood"), and mass. The functional tier defines part-level attributes such as intrinsic function (e.g., "to contain") and state machines (e.g., [open, closed]). Finally, the interactive tier specifies kinematic properties, including joint types (e.g., revolute, prismatic), and atomic affordances (e.g., pushable, graspable).

PhysForge ultimately achieves the generation of functionally complete, physically interactive 3D assets from a single view image. Extensive experiments and qualitative demonstrations in physics simulators and game virtual worlds validate the effectiveness of the method, providing high-fidelity, interactive assets for downstream applications such as robotic manipulation and game development.

Our core contributions are summarized as follows:

- Formulation and Framework: We propose a novel formulation for physics-grounded 3D generation, and a decoupled *VLM-based Planning + Diffusion-based Generation* two-stage framework (PhysForge).

- Large-scale Dataset: We contribute a large-scale, part-aware dataset with fine-grained, physical annotations (PhysDB), filling a critical data gap in the field.

- Extensive Validation and Application: We provide extensive experiments validating our framework's state-of-the-art performance on both planning and generation, and demonstrate the direct applicability of our assets in robotic simulators and interactive virtual worlds.

## 2. Related Work

### 2.1. 3D Content Generation

The field of 3D content generation has rapidly expanded, largely following two distinct philosophies: leveraging powerful 2D priors or training directly on 3D data. A foundational strategy, Score Distillation Sampling (SDS) pioneered by DreamFusion (Poole et al., 2023), enables text-to-3D synthesis without 3D supervision by optimizing a 3D representation using gradients from a 2D model. Significant progress has been made on 3D-native generation, with models such as 3DShape2VecSet (Zhang et al., 2023) introducing an encoding scheme that uses cross-attention for set-structured 3D data, CLAY (Zhang et al., 2024) scaling 3D diffusion to massive datasets, and TRELLIS (Xiang et al., 2024) introducing structured latents for a high-quality, coarse-to-fine generation process. Despite this rapid evolution in synthesizing high-fidelity geometry and textures, a common limitation unites all these approaches: the resulting assets are holistic and non-interactive.

### 2.2. Part-aware 3D Shape Generation

Recognizing the limitations of holistic generation, a recent line of work has begun to explore part-aware 3D generation. The central challenge in this sub-field is how to decompose a complex object into meaningful components while ensuring the final structure remains geometrically coherent. Early approaches have primarily adopted one of two strategies. The first is a "reconstruction-from-views" pipeline, which leverages 2D part masks to guide multi-view reconstruction (Liu et al., 2024a; Chen et al., 2024a). A significant advancement came from OmniPart (Yang et al., 2025), which introduced a two-stage framework built upon TRELLIS (Xiang et al., 2024) to achieve semantic decoupling and structural cohesion, enabling controllable part generation. Other approaches, like PartPacker (Tang et al., 2025), have focused on representation efficiency, compressing all parts into a compact dual volume representation for efficient generation from a single image. Critically, all these methods define *parts* based on purely geometric or visual boundaries. Their goal is to create assets that are visually decomposable. This leaves a crucial gap: the function and physics of a part are never considered.

### 2.3. Physics Grounded 3D Shape Generation

Recently, a few pioneering works have begun to bridge the gap between static geometry and interactive physics. PhysX-3D (Cao et al., 2025) makes a significant contribution by introducing PhysXNet, a dataset annotating physical properties on top of PartNet (Mo et al., 2019), and a generation model based on TRELLIS (Xiang et al., 2024) using a Physical VAE. Separate from holistic physics, another body of research has focused specifically on articulation, a key component of interaction. This research has diverged into two main directions. One specialized direction has concentrated on the reconstruction of articulated objects, often termed "Digital Twins" (Liu et al., 2023; 2025b; Weng et al., 2024; Wu et al., 2025; Song et al., 2024; Tu et al., 2025; Shen et al., 2025). A second direction attempts procedural generation of articulated assets (Chen et al., 2024b; Gao et al., 2025; Le et al., 2024; Liu et al., 2024c;b; Mandi et al., 2024; Qiu et al., 2025). These approaches often rely on external, predefined content, such as part repositories, code templates, or VLM-predicted connectivity graphs, which constrains their ability to generalize to novel object categories and often leads to suboptimal accuracy.

## 3. Physics-Grounded, Part-Aware 3D Assets Generation

Our goal is to generate physics-grounded 3D assets that can serve a wide range of domains, from embodied AI simulation environments to interactive video games. To achieve this, our approach is built upon two pillars: (1) a comprehensive and diverse training dataset, and (2) a powerful and robust generation pipeline. We first introduce PhysDB, a novel large-scale dataset, in Section 3.1. It provides rich, fine-grained physical annotations necessary for this task. Following this, we introduce an innovative two-stage generation framework, PhysForge, as shown in Figure 2. Stage 1 (Section 3.2) is a "VLM Planner" that generates a hierarchical physical blueprint. Stage 2 (Section 3.3) is a "Diffusion Realization" stage, which uses a novel KineVoxel Injection mechanism to synthesize high-fidelity geometry, texture, and precise articulation parameters.

### 3.1. PhysDB: A Physics-Grounded Dataset

We propose a system of annotation that uses **holistic**, **static**, **functional**, and **interactive properties** to define the physical nature of each asset. At the object level, we define the asset's real-world scale, its object category, and its intended usage scene (e.g., kitchen, bedroom). Descending to the part level, we first define static and semantic properties, such as the part's semantic label, its physical material, and its mass. Next, we define functional properties inspired by OAKINK2 (Zhan et al., 2024), which include the part's intrinsic function (e.g., "to contain", "to control") and its potential state machine (e.g., Button: [pressed, released]). Finally, our interactive tier specifies how an agent can interact with the object, detailing an atomic affordance library (e.g., pushable, rotatable) and, for movable parts, their complete kinematic definition: a parent part, a joint type (revolute, continuous, prismatic, or fixed), and the precise joint parameters (axis origin, direction, and limits).

We introduce PhysDB, a new dataset of 150k 3D objects

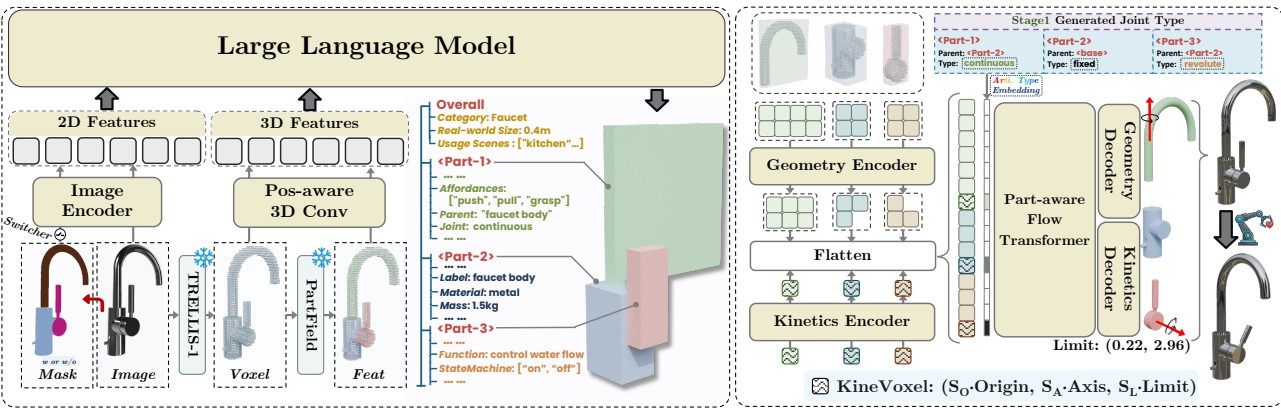

*Figure 2.* **Method overview.** PhysForge consists of two stages: (Left) Stage 1: VLM-based Planning, where the VLM planner generates a "Hierarchical Physical Blueprint" defining part structure and physical properties. (Right) Stage 2: Diffusion-based Generation, where a diffusion model, guided by the blueprint, uses the KineVoxel Injection (KVI) mechanism to synergistically generate the final geometry, texture, and precise kinematic parameters.

sourced from Objaverse (Deitke et al., 2023), covering seven major categories: household, industrial, weapons, personal, vehicles, tech & electronics, and cultural items. We select objects that are amenable to our physics annotation pipeline and already possess a meaningful part structure. Our annotation pipeline involves a human-in-the-loop process. We first render whole objects and per-part images, which are fed to a multimodal LLM to generate initial annotations. This is followed by manual screening and correction to ensure the accuracy and consistency of the final PhysDB dataset. Scaling precise 3D articulation annotation to 150k objects is extremely challenging. Due to the wide variety of object categories, PhysDB focuses on providing rich physical properties and identifying joint types, rather than attempting to annotate precise numerical axes which are often inaccurate at this scale. To bridge this kinematic gap, we supplement our training process with PartNet-Mobility (Xiang et al., 2020) and Infinite-Mobility (Lian et al., 2025), which provide the ground-truth articulation parameters necessary to train our model in the diffusion stage.

## 3.2. VLM as a Physical Blueprint Planner

The VLM's rich world knowledge provides a strong prior for object-part relationships, making it an ideal *planner* for our first stage. While VLMs lack explicit 3D understanding, we finetune them to evoke this capability. We select Qwen2.5-VL (Bai et al., 2025) as our base model due to its powerful knowledge base and vision capabilities. To integrate 3D information, the model accepts a single image $I$, its corresponding 3D voxel representation $V$ (obtained from TRELLIS (Xiang et al., 2024) first stage), and an optional 2D part mask $M$ for granularity control. The input image $I$ and the 2D mask $M$ (which is converted to a color map) are processed directly by Qwen's powerful image encoder. For the 3D voxel input $V$, we diverge from the common

3DShape2VecSet (Zhang et al., 2023) encoder. To better capture part-aware and local information, we first use a Part-Field encoder (Liu et al., 2025a) to extract features for each voxel, then apply a position-aware 3D convolutional network to downsample these features into a 512-dimensional voxel embedding.

With these encoded inputs, we finetune the VLM to autoregressively generate the complete part structure and physical properties. We introduce 66 new special tokens to the VLM's codebook: <boxs> and <boxe> to delimit a bounding box, and 64 discrete tokens (<box0>, ..., <box63>) for the quantized coordinates. Each 3D axis-aligned bounding box is thus represented by only 6 tokens, enabling highly efficient structural planning. The model then outputs the hierarchical physical blueprint for each planned part. A key discovery is that physics-guided planning resolves part ambiguity. Training the model to co-predict physical properties (like material and function) alongside bounding boxes provides stronger semantic constraints. This synergy significantly improves the model's understanding of part decomposition. As a result, even when no 2D mask is provided, the VLM can produce semantically coherent and reasonable bounding box plans.

## 3.3. Diffusion-based Generation with KineVoxel Injection

The VLM planner outputs a hierarchical structure, including per-part bounding boxes, parent-child relationships, and semantic joint types (e.g., fixed, revolute). While the VLM excels at this high-level structural and semantic planning, it is ill-suited for predicting the precise, continuous 3D values required for kinematics, such as an exact origin coordinate or axis vector. We therefore delegate this task to the diffusion head. This presents a challenge: how to synergistically generate these continuous parameters within a diffusion

*Table 1.* Quantitative comparison of Physics Property generation on the PhysXNet. Our method outperforms the baseline in both geometric generation quality and the accuracy of predicted physical properties.

| Method | CD ↓ | F1-0.1 ↑ | F1-0.05 ↑ | Absolute scale (cm) ↓ | Material ↓ | Affordance ↓ | Description ↑ |
|---|---|---|---|---|---|---|---|
| TRELLIS | 10.10 | 86.53 | 72.47 | - | - | - | - |
| PhysXGen | 9.81 | 87.91 | 73.60 | 25.83 | 1.59 | 3.69 | 0.38 |
| PhysForge (Ours) | **9.21** | **89.24** | **75.43** | **11.04** | **0.81** | **1.22** | **0.87** |

*Table 2.* Quantitative comparison of Physics Property generation on the PhysDB. On our more diverse PhysDB dataset, our model demonstrates a more significant advantage over the baseline methods.

| Method | CD ↓ | F1-0.1 ↑ | F1-0.05 ↑ | Absolute scale (m) ↓ | Material ↓ | Function ↑ | Interaction ↑ |
|---|---|---|---|---|---|---|---|
| TRELLIS | 24.32 | 68.19 | 53.28 | - | - | - | - |
| PhysXGen | 25.30 | 65.79 | 50.57 | 1.08 | 1.44 | 0.36 | 0.34 |
| PhysForge (Ours) | **22.89** | **70.51** | **55.38** | **0.37** | **0.43** | **0.83** | **0.96** |

*Table 3.* Quantitative results for bounding box generation (%) on PartObjaverse-Tiny. Results show that physics-guided planning significantly improves part planning accuracy and enables semantically reasonable results even without a 2D mask input.

| Method | Voxel recall ↑ | Voxel IoU ↑ | Bbox IoU ↑ |
|---|---|---|---|
| PartField | 69.65 | 46.04 | 37.33 |
| OmniPart (SAM mask) | 68.33 | 43.34 | 34.33 |
| PhysForge-bbox (w/o mask) | 67.89 | 35.53 | 32.30 |
| PhysForge (w/o mask) | 73.63 | 47.66 | 36.32 |
| OmniPart | 73.79 | 52.92 | 41.66 |
| PhysForge (Ours) | **77.16** | **53.74** | **42.95** |

pipeline designed for geometry?

We solve this by extending the OmniPart (Yang et al., 2025) second stage framework with our novel KineVoxel Injection mechanism. Our approach begins by representing the articulation parameters for a single part $i$ as an 8-dimensional vector $P_i = (O_i, A_i, L_i)$, where $O_i \in \mathbb{R}^3$ is the joint origin, $A_i \in \mathbb{R}^3$ is the joint axis, and $L_i \in \mathbb{R}^2$ is the motion limits. We represent $P_i$ as a "KineVoxel", a special representation that can be processed alongside the standard geometric latents $Z_g$ in a unified denoising framework. Our approach maps data from different modalities (geometry and kinematics) into a unified latent space for joint diffusion. We utilize independent Kinematic Encoders ($E_{kine}$) and Decoders ($D_{kine}$) to process the KineVoxel, allowing it to share a latent space with the geometry latents within the middle transformer:

$$z_{k,i} = E_{kine}(\text{concat}(S_O \cdot O_i, S_A \cdot A_i, S_L \cdot L_i)),$$

where $S_O, S_A, S_L$ are scaling factors. Both $E_{kine}$ and $D_{kine}$ are implemented as lightweight 2-layer MLPs. The diffusion network contains down-sample blocks, a middle transformer, and up-sample blocks. We inject our KineVoxel $z_{k,i}$ after downsampling, concatenating it with the sequence of geometry voxel latents $Z_g = \{z_{g,i}\}$ before they are fed into the main denoising transformer. To allow the transformer to distinguish between the two latent types, we add a joint type embedding $E_{type}$ to the KineVoxel. This embedding $E_{type}$ is derived from the VLM's planned joint type (e.g., "revolute") and is added to $z_{k,i}$. The transformer can thus learn the complex correlations between part geometry

and its corresponding joint parameters.

The entire model is trained by minimizing the Conditional Flow Matching (CFM) objective (Lipman et al., 2024). We define a composite loss that separates the contribution of geometry and kinematic voxels:

$$\mathcal{L} = \mathbb{E}_{t, Z_0, c} [\mathcal{L}_{geo} + \lambda_{kine} \cdot \mathcal{L}_{kine}]$$

where $c$ is the condition from the VLM blueprint. The loss terms $\mathcal{L}_{geo}$ and $\mathcal{L}_{kine}$ are the standard $L_2$ losses between the predicted and target velocities for the geometry latents $Z_g$ and kinematic latents $Z_k$, respectively:

$$\mathcal{L}_{geo} = \|v_{g,t} - \hat{v}_{g,t}\|^2; \mathcal{L}_{kine} = \|v_{k,t} - \hat{v}_{k,t}\|^2.$$

We set the weighting factor $\lambda_{kine} = 10$ throughout our training, placing a higher importance on accurately predicting the precise articulation parameters.

## 4. Experiments

**Evaluation Protocol.** To evaluate our model, we utilize the commonly used part-level dataset PartObjaverse-Tiny (Yang et al., 2024), which contains 200 diverse objects, and the test set (1000 objects) from PhysXNet (Cao et al., 2025). We also establish two new test sets: (1) a set of 1,000 cases sampled uniformly by category from our proposed PhysDB, and (2) a set of 340 articulated objects sampled from PartNet-Mobility and Infinite-Mobility. We first evaluate our model's capability in the "Part Structure Planning via VLM" stage on the PartObjaverse-Tiny dataset, with results presented in Section 4.1. Following this, in Section 4.2, we evaluate the model's performance on generating accurate physical properties and kinematic parameters. Finally, we demonstrate the broad applications of our model in Figure 6.

### 4.1. Part Structure Planning

**Baselines and Metrics.** We first evaluate and analyze our model's capability on the Part Structure Planning task. We select the first stage of OmniPart (Yang et al., 2025) and

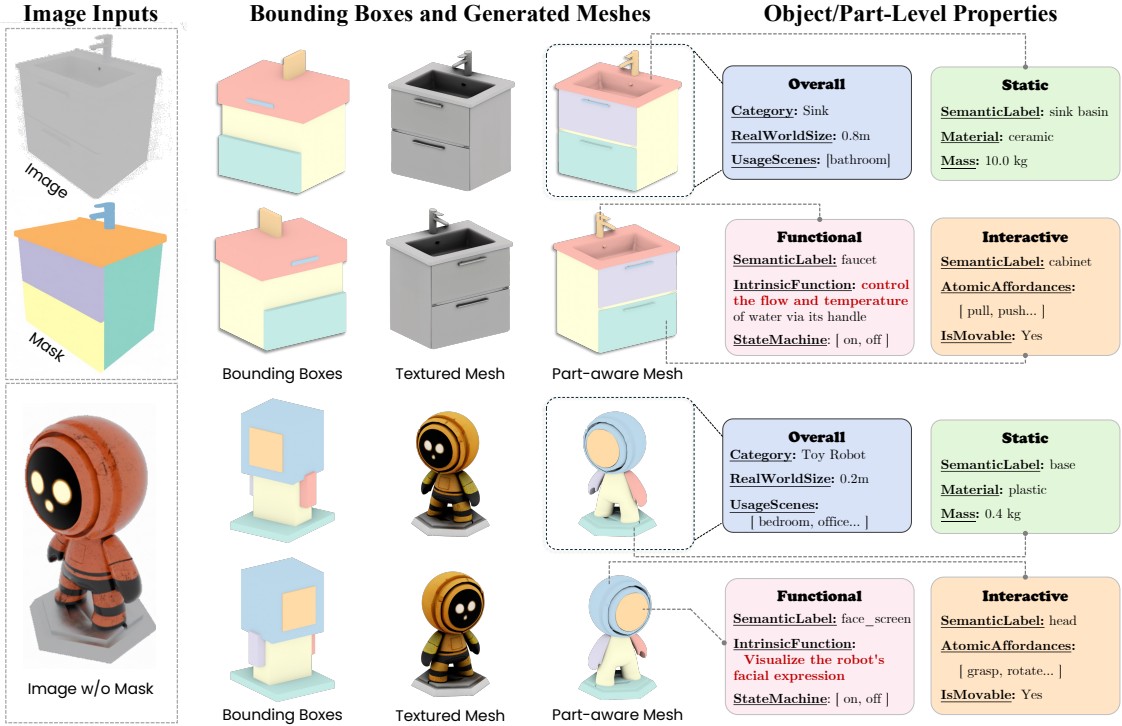

*Figure 3.* Qualitative results of PhysForge. Given a single image and an optional 2D mask for control, our model generates high-quality, physics-grounded, and part-aware 3D assets.

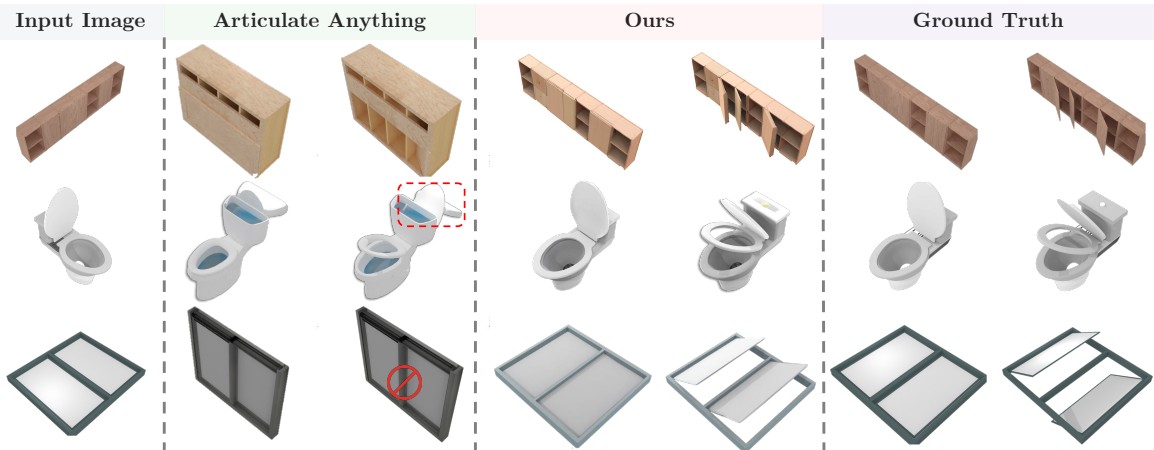

*Figure 4.* Qualitative results of articulated object generation from a single image.

PartField (Liu et al., 2025a) as our primary baselines. The first stage of OmniPart trains an auto-regressive transformer on part-level data for bounding box generation, which, by default, requires a 2D mask input to control the granularity of the generated parts. PartField is a point cloud segmentation method that can also take voxels as input to produce voxel-level segmentation results and corresponding bounding boxes. As PartField requires the number of parts to define the segmentation scale, we provide the ground-truth number of parts as input. Following OmniPart, we use BBox IoU, Voxel Recall, and Voxel IoU as our evaluation metrics,

assessing both bounding box-level accuracy and voxel-level planning precision.

**Results and Ablation Analysis.** In Table 3, we show the comparison of all methods on the part structure planning task. To analyze our model's planning ability with and without a 2D mask input, we introduce two additional experimental settings. The second row, "OmniPart (SAM mask)", replaces OmniPart's ground-truth 2D mask with a 2D mask obtained from SAM (Kirillov et al., 2023), filtering out small masks with an area ratio less than $1600/1024^2$. The third

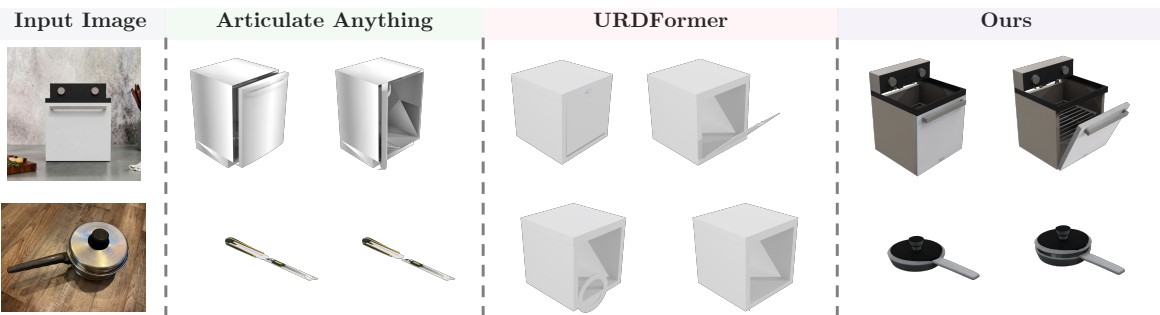

*Figure 5.* Qualitative results of articulated object generation from an in-the-wild image.

row, "PhysForge-bbox", represents our model architecture trained only on the 500k part-level bounding box dataset (without physics). An entry marked "w/o mask" indicates that no mask was provided to the model input.

Comparing the overall results, our full model achieves state-of-the-art results, demonstrating the strongest part structure planning capability. The results of "PhysForge w/o mask" (row 4) are significantly better than the "PhysForge-bbox" model (row 3), which demonstrates that the introduction of physical properties significantly enhances our model's semantic understanding and planning capabilities for part structures. Even without a mask input, it can still produce semantically reasonable results. Furthermore, our model operating without a mask still outperforms the OmniPart's first stage that uses SAM-generated masks, highlighting the robustness of our physics-guided planning.

### 4.2. Physics-Grounded Generation

**Baselines and Metrics.** We evaluate our model's ability to generate physics properties by selecting PhysXGen and TRELLIS as our primary baseline. Specifically, we normalize the ground-truth and predicted shapes into a canonical space of [-0.5,0.5], then compute the Chamfer Distance (CD) and F1-Score. The F1-Score is assessed at two distance thresholds, $CD < 0.1$ and $CD < 0.05$. To evaluate the accuracy of physics properties at the part level, we compare the MAE of Absolute Scale, Material, Affordance, and the CLIP-Similarity of text-based Function and Interaction.

To evaluate our model's performance in generating Kinematic Parameters, we select Articulate Anything (Le et al., 2024), Singapo and URDFormer as baselines, as they support articulated object generation from a single image. For this task, we use CD (%) and F-Score (%) to measure mesh generation quality, and CLIP-Similarity to assess the match with the input image. Following Articulate Anything, we utilize Joint Axis Error and Joint Pivot Error to measure the accuracy of the generated kinematic parameters. Specifically, we report Joint-Axis-Err-5 and Joint-Pivot-Err-5 on the subset of categories supported by all methods, and additionally report Joint-Axis-Err-all and Joint-Pivot-Err-all for

methods that generalize to all categories.

**Physics Properties.** In Table 1, we present a comparison of our method against the baselines PhysXGen and TREL-LIS. Our method surpasses the other methods in terms of geometry generation quality. Unlike PhysXGen, which is trained on specific categories and is limited to outputting opaque CLIP features, our method benefits from the VLM's powerful world-knowledge prior, enabling it to directly and accurately output corresponding physics properties as both text and numerical values. Therefore, in the realm of physics property generation, our method significantly outperforms the baseline. Furthermore, in Figure 3, we demonstrate our model's effectiveness in generating physics-grounded assets. From a single image, along with optional 2D mask control, our pipeline can accurately plan all part bounding boxes and physical attributes, and subsequently utilize the diffusion model to generate part-aware geometry and textures.

**Kinematic Parameters.** In Table 4, we present the quantitative comparison between our method and the baseline models, along with qualitative results on the validation set (Figure 4) and on in-the-wild images (Figure 5). The articulated objects generated by our method are significantly superior to the baseline in terms of both consistency with the input image and the accuracy of the joint parameters.

**Ablation Analysis.** We report the results of two key ablation studies in Table 4, which analyze the impact of removing the joint type embedding and the dedicated kinematic sub-network. The joint type embedding serves as the critical interface between our two stages: while Stage 1 predicts the qualitative articulation type (e.g., revolute, prismatic), this embedding provides a strong functional prior that constrains and guides the precise parameter estimation in Stage 2. The results clearly demonstrate that without the guidance from Stage 1's planning, Stage 2 struggles to resolve kinematic ambiguities, leading to a degradation in joint accuracy, confirming that the joint type embedding is indispensable for effectively transferring physical common sense to the generation stage. Furthermore, removing the independent kinematic encoder and decoder further compromises the model's ability to synthesize precise mechanical constraints.

*Table 4.* Quantitative comparison of articulated objects generation. Our method achieves higher fidelity to the input image and more accurate joint axis and pivot prediction.

| Method | CD ↓ | CLIP-Sim ↑ | Joint-Axis-Err-5 ↓ | Joint-Pivot-Err-5 ↓ | Joint-Axis-Err-all ↓ | Joint-Pivot-Err-all ↓ |
|---|---|---|---|---|---|---|
| Articulate Anything | 23.31 | 0.87 | 0.608 | 0.257 | 0.694 | 0.197 |
| Singapo | 21.10 | 0.85 | 0.241 | 0.153 | - | - |
| URDFormer | 25.42 | 0.84 | 0.781 | 0.652 | - | - |
| PhysForge (w/o joint type emb) | 10.73 | 0.90 | 0.157 | 0.132 | 0.292 | 0.141 |
| PhysForge (w/o kinematic enc) | 11.31 | 0.89 | 0.158 | 0.117 | 0.204 | 0.120 |
| **PhysForge** (Ours) | **10.21** | **0.93** | **0.101** | **0.071** | **0.164** | **0.096** |

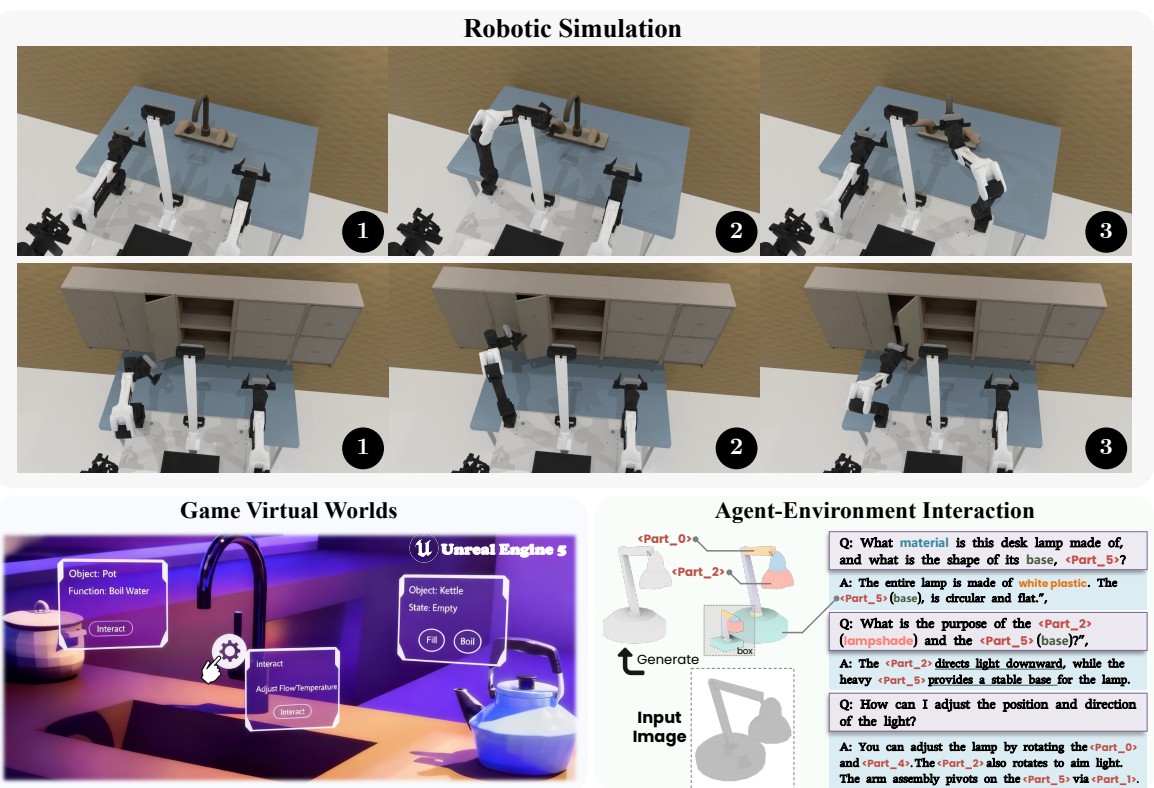

*Figure 6.* **Downstream Applications of PhysForge.** Our generated assets are simulation-ready: (a) A robotic arm manipulates an asset's functional parts in a RoboTwin (Mu et al., 2025; Chen et al., 2025b) simulator. (b) The assets are imported into a virtual world (e.g., Unity/UE), enabling rich, physics-based interactions. (c) An agent interacts with our model via natural language, querying its physical blueprint to plan a task.

## 4.3. Application

To demonstrate the downstream utility of our generated assets, we showcase three primary applications in Figure 6: (a) Robotic Simulation: We demonstrate that our generated assets can be successfully imported into the RoboTwin (Mu et al., 2025; Chen et al., 2025b) simulation environment. The detailed part-level geometry and precise kinematic parameters allow robotic manipulators to realistically interact with the objects. (b) Virtual Worlds: In game engines and virtual worlds, our assets enable complex interactions. Because every part is endowed with physics-grounded attributes (materials, mass, articulation), developers can design sophisticated interaction logic without manual rigging. (c) Agent-Environment Interaction: Our VLM-based framework opens a new modality for interaction. An embodied agent (or VLA) can directly query our model in natural

language and receive a text-based physical blueprint with bounding boxes, providing an explicit plan for manipulation.

## 5. Conclusion

We introduce PhysForge, a novel framework that generates interactive and physics-grounded 3D assets. Our decoupled "VLM Planning + Diffusion Realization" architecture fine-tunes a VLM to generate "Hierarchical Physical Blueprints" that define an asset's complete physical profile. To realize these blueprints, our KineVoxel Injection algorithm enables a diffusion model to synergistically generate geometry and precise kinematic parameters. This framework is supported by PhysDB, our large-scale, 150k-asset dataset with rich annotations. PhysForge provides a foundational data engine for embodied AI and interactive virtual worlds.

## Acknowledgements

This work is partially supported by the National Nature Science Foundation of China (No. 62402406).

## Impact Statement

This paper presents work whose goal is to advance the field of machine learning. There are many potential societal consequences of our work, none of which we feel must be specifically highlighted here.

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
