# OpenReview forum: "PhysForge: Generating Physics-Grounded 3D Assets for Interactive Virtual World"
_ICML.cc/2026/Conference — ICML 2026 regular_

### Official Review · Reviewer_S8xF · 2026-02-27

**Soundness:** 3
**Presentation:** 2
**Significance:** 3
**Originality:** 2
**Overall Recommendation:** 4
**Confidence:** 4

**Summary:**

The paper introduces PhysForge, a two-stage framework for generating physics-grounded, interactive 3D assets from single images. The framework adopts a planning-then-generation paradigm. In the first stage, a VLM (Qwen2.5-VL) acts as a planner to predict a hierarchical physical blueprint, including part bounding boxes and physical properties. In the second stage, a diffusion model integrates these blueprints using a novel KineVoxel Injection (KVI) mechanism to jointly synthesize high-fidelity geometry and precise articulation parameters. To support this, the authors introduce PhysDB, a large-scale dataset of 150k assets with a 4-tier physical annotation system.

**Compliance With Llm Reviewing Policy:**

Affirmed.

**Final Justification:**

This paper tackles an important and timely problem: generating physics-grounded, interactive 3D assets from a single image. I find the overall idea compelling, and the combination of a VLM-based planner, a diffusion-based generation stage, and a large-scale annotated dataset makes this a meaningful systems contribution. The strongest technical component is the articulation-aware generation design in Stage 2, and overall I find the work technically solid and promising.

The rebuttal addressed a meaningful portion of my concerns. In particular, it clarified several evaluation details, added helpful ablations, and made the role of the different components easier to understand. My main remaining concern is that the paper should more clearly distinguish the role of PhysDB from the role of external mobility datasets in Stage 2. As I understand it after the rebuttal, PhysDB mainly provides large-scale physical/functional supervision and geometry regularization, while precise continuous kinematic supervision is supplemented by external mobility datasets. I also think the part-level property evaluation protocol should be described more explicitly for transparency.

Overall, I agree with the direction and contribution of the work. With these clarifications in the revised version, I am comfortable updating my final recommendation to Weak Accept.

**Key Questions For Authors:**

1. Data Alignment: How exactly was the part alignment performed between the Objaverse-based PhysDB and the kinematic parameters from PartNet-Mobility? If the kinematic training relies solely on the latter, could you clarify the exact contribution of PhysDB in the diffusion stage?
2. Ablation on Annotations: Can you provide an ablation study demonstrating how the specific tiers of PhysDB (particularly the functional and interactive tiers) individually contribute to the final generation quality?
3. Data Leakage and Generalization: How did you account for potential data leakage from Qwen2.5-VL’s pre-training? Could you provide zero-shot evaluation results on completely out-of-distribution (OOD) objects to rule out memorization?
4. Attention Mechanism: Could you provide attention map visualizations between the dense geometry voxel tensor and the sparse KineVoxel tensor to prove that meaningful cross-talk is occurring during the diffusion process?
5. Concurrent Work: How does PhysForge fundamentally differ in architecture and capability from concurrent SOTA like PhysX-Anything (CVPR 2026)?

**Limitations:**

The following points should be explicitly addressed in a revised Limitations section:
- "Simulation-Ready" Claims: The authors heavily claim the assets are simulation-ready, yet they ignore the severe bottleneck of mesh penetration and convex decomposition required for actual physics engines . The gap between a raw diffusion mesh and a functional simulation asset must be acknowledged.
- Cascading Errors: The framework is highly susceptible to cascading errors. If the single-image VLM planner hallucinates incorrect materials or joints, the downstream diffusion model is forced to generate flawed geometry. The lack of robustness against these two-stage cascading failures must be discussed.
- Dataset Transparency: The authors should be upfront about the fact that PhysDB lacks continuous kinematic values, necessitating reliance on prior mobility datasets.

**Strengths And Weaknesses:**

# Strengths
The paper tackles a highly relevant and challenging problem: moving beyond static 3D generation to create functionally interactive, physics-grounded assets. The ambition to construct a large-scale physical dataset (PhysDB) and the integration of a VLM planner with a diffusion-based generator are commendable. The proposed KVI mechanism is an interesting approach to unifying geometric and kinematic generation.

# Weaknesses
### Originality & Significance:
While the authors claim the two-stage (planning-then-generation) framework as a major contribution, this pipeline itself is somewhat standard in recent literature. The true methodological novelty lies in the KVI mechanism, which should be the central focus. Furthermore, the paper misses a critical comparison with concurrent SOTA work. Specifically, PhysX-Anything (CVPR 2026) closely shares the goal of unified geometry, articulation, and physical attribute generation. Acknowledging this concurrent work and clarifying PhysForge’s unique architectural advantages is necessary to establish its originality.

### Soundness (Methodology & Experiments):
I have several concerns regarding the technical soundness of the data processing, architectural design, and evaluation protocol:
- Dataset Originality & Alignment: The authors heavily promote PhysDB (150k assets) and its 4-tier annotations. However, the most critical data for training KineVoxels—continuous kinematic parameters—is absent from PhysDB and relies entirely on external datasets (PartNet-Mobility, Infinite-Mobility). It is unclear how the authors performed part alignment between the Objaverse-sourced PhysDB and PartNet-Mobility . If the diffusion model relies strictly on external datasets for kinematics, the contribution of PhysDB is somewhat overstated. Furthermore, there is no ablation study proving that the functional/interactive tiers of the annotations actually improve generation performance. Also, the error rates and human-validation details for the VLM-automated annotations are missing.
- VLM Planner Constraints: The planner quantizes 3D space into 64 discrete tokens. While sufficient for macro-structures, this aggressive quantization likely causes severe precision errors for fine-grained parts (e.g., small hinges). Additionally, inferring intrinsic properties (mass, material) from a single 2D image is inherently an ill-posed problem; without handling failure cases, it is hard to verify if the VLM is truly "physics-grounded" or merely hallucinating based on prior statistics. The computational efficiency of embedding heavy 3D encoders (TRELLIS + Part Field Encoder) into the VLM planner is also not discussed or justified via IoU ablations.
- Physics-Grounded Generation: Training the KineVoxel with a simple L2 loss does not guarantee physical validity. In 3D dynamics, physical plausibility requires strict avoidance of inter-part collisions and self-penetration . The lack of a kinematic collision penalty is a significant oversight. Furthermore, it is not intuitively clear how dense geometry voxels and sparse KineVoxels attend to each other effectively; visualizing the attention maps would make this much more convincing.
- Evaluation Protocol: The evaluation heavily favors the proposed method. First, using Qwen2.5-VL raises data leakage concerns, as the test sets (PartObjaverse, PartNet-Mobility) were likely seen during the VLM's pre-training. Second, testing mostly on in-distribution samples without a strict Zero-shot/OOD evaluation makes it difficult to assess true reasoning capability versus memorization—especially in Table 4, where the test objects share the domain of the finetuning data. Comparing the VLM planner to a lightweight OmniPart baseline is unfair due to the massive difference in model capacity. Finally, comparing text similarities against PhysXGen (which outputs CLIP features, not text) is not a meaningful metric.

### Presentation:
The narrative is generally clear, but there are several disruptive formatting and writing errors that need addressing.
- Typos: "multi modal" should be "multimodal", and "Kinetics" should be corrected to "Kinematics" in Figure 2 and Table 4.
- In the Evaluation Protocol section, the final sentence incorrectly references Section 4.2 instead of 4.3.
- Table referencing is disjointed: Table 3 is referenced before Tables 1 and 2. Tables 2, 5, and 6 are not mentioned in the text at all, and there is an unresolved placeholder [Yunhan: table] that seems to point to Table 5.

---

> ### Author Rebuttal · Authors · 2026-03-31
>
> Thank you very much for taking the time to review and for your support. We try our best to address your questions as follows.
>
> **Originality & Significance.** While the planning-then-generation paradigm is common, applying it to physics-aware 3D generation remains novel.
> We fundamentally differ from the concurrent PhysX-Anything. It forces a VLM to directly output continuous kinematic values and low-resolution voxels, inherently struggling with precise 3D spatial positioning. It then extracts parts by matching mesh faces to these coarse voxels, making accurate surface-level segmentation difficult. In contrast, PhysForge is part-based from the ground up. We restrict the VLM to the semantic attributes it excels at, while delegating precise numerical kinematics and independent part mesh generation to a 3D-aware diffusion head. We have provided a qualitative visual comparison with PhysX-Anything in the attached [anonymous link](https://postimg.cc/k2w9xQK8) to clearly demonstrate these structural advantages.
>
> **PhysDB.** PhysDB’s massive scale (150k assets) and diversity are crucial for our framework. Our hybrid training strategy utilizes PhysDB to train both the Stage 1 VLM for physical reasoning and the Stage 2 diffusion model for mesh generation. To provide continuous kinematics, we supplement Stage 2 with data from PartNet-Mobility and Infinite-Mobility. This prevents overfitting to limited mobility datasets, enabling high-fidelity, part-aware generation for diverse in-the-wild assets (Figs. 3 & 8). Due to space constraints, please see our response to Reviewer PiE5 for dataset construction details. Additionally, screenshots of our annotation system are provided in the attached [anonymous link](https://postimg.cc/jC9Q8yqb).
>
> **VLM.**
> - The 64-resolution is highly efficient for Stage 1 part planning. Stage 2 upscales this to 256 for high-fidelity geometry without compromising mesh quality.
> - Though a single image is incomplete, PhysDB fine-tuning equips our VLM with strong priors to identify visible materials and infer properties of occluded regions.
> - TRELLIS only initializes the 64-resolution voxels for our lightweight PartField Encoder. The ablation below shows it efficiently captures richer part-level semantics than standard vecset encoders.
>
> |Encoder|Voxel recall|Voxel IoU|Bbox IoU|
> |-|-|-|-|
> |Vecset|70.61|46.89|37.21|
> |PartField(ours)|**77.16**|**53.74**|**42.95**|
>
> **Physics-Grounded Generation.**
> - To prevent self-penetration, our model uses part embeddings to explicitly distinguish adjacent regions and mitigate inter-part collisions during generation. Additionally, an automated verification step directly adjusts the predicted kinematic limits before simulation to eliminate any motion-induced self-intersections.
> - Attention Maps: Visualizations in the attached [anonymous link](https://postimg.cc/KKG5nRRW) demonstrate that the activated regions of sparse KineVoxels strongly correlate with the dense geometry of their corresponding parts. This confirms that continuous KineVoxel predictions effectively attend to the overall structural mesh information throughout the diffusion process.
>
> **Evaluation Protocol.**
> - We strictly excluded all test sets during VLM fine-tuning. We view the VLM's extensive world knowledge as a core advantage rather than a confounding variable, making it highly practical to leverage this robust prior for physical reasoning. Additionally, Figure 5 demonstrates strong in-the-wild generalization, and we welcome suggestions for specific zero-shot or OOD benchmarks.
> - We compared against OmniPart and PartField because they address identical part-level planning tasks. While model capacities naturally differ, effectively harnessing a stronger VLM prior is precisely our framework's architectural advantage.
> - Because PhysXGen outputs CLIP features instead of natural language, we evaluated our method by extracting the CLIP features of our generated text descriptions. The protocol is also adopted by the concurrent work PhysX-Anything.
>
> **Typos.** We will carefully correct all of them in the revised manuscript.
>
> **Ablation on Annotations.** Incorporating rich physical attributes significantly improves the VLM's physical reasoning during part planning. The ablation study below quantitatively evaluates part planning performance when they are independently removed.
>
> |Method|Voxel recall|Voxel IoU|Bbox IoU|
> |-|-|-|-|
> |Only-bbox (w/o mask)|67.89|35.53|32.30|
> |W/o func|71.38|43.81|33.91|
> |W/o inter|72.17|45.82|35.42|
> |W/o mask|**73.63**|**47.66**|**36.32**|
>
> **Limitations**
> - Acknowledging the gap between raw meshes and simulation assets requiring post-processing, though Figure 6 proves practical viability.
> - Incorrect VLM joint type predictions propagate to the diffusion model, whereas material errors do not degrade geometry.
> - Efficiently scaling precise kinematic annotations to PhysDB's massive size remains future work.

---

> > ### Author Rebuttal · Reviewer_S8xF · 2026-04-03
> >
> > I carefully read the rebuttal and additional materials. My current assessment is (b) Partially resolved, and I keep my score unchanged for now.
> >
> > The rebuttal addressed several earlier concerns in a useful way. In particular, the authors clarified the architectural distinction from concurrent work, added an ablation showing that richer functional/interactive annotations improve Stage-1 planning, and acknowledged important limitations more explicitly. These points improve the paper.
> >
> > However, the main soundness and evaluation concerns are not yet fully resolved. Most importantly, the exact role of PhysDB in Stage 2 remains unclear: the rebuttal now makes clear that precise continuous kinematics are mainly supplemented by external mobility datasets, but it does not clearly specify what PhysDB contributes to Stage 2 beyond general part-aware generation. The rebuttal also introduces an automated verification step for adjusting kinematic limits, but it is unclear whether the Table 4 articulated-object metrics are measured before or after this post-processing, and whether comparable processing is applied to baselines. In addition, Appendix A still appears to use asymmetric property-matching procedures across methods for part-level evaluation, which may affect fairness.
> >
> > The annotation-pipeline and generalization discussions are more informative after the rebuttal, but they remain only partially addressed and do not materially change my assessment.
> >
> > Overall, the rebuttal improved the paper, but it did not fully resolve the key concerns that drove my original score. I therefore keep my score unchanged and asked the authors for focused clarification on the points above.

---

> > > ### Author Response · Authors · 2026-04-07
> > >
> > > We sincerely thank the reviewer for acknowledging our previous revisions and for raising these constructive follow-up questions.
> > >
> > > **The Exact Role of PhysDB.**
> > >
> > > Its critical role spans both stages:
> > >
> > > Stage 1 (Physics-Aware Planning): Teaching a VLM complex physical reasoning requires more than just architectural design; it heavily relies on massive 3D data featuring broad category diversity and rich, multi-tier physical annotations. PhysDB serves as the irreplaceable knowledge foundation for this fine-tuning, directly endowing the VLM with its core physics-aware capabilities. Furthermore, only with such a massive and diverse dataset can we effectively implement 3D voxel augmentations (e.g., random scaling and erosion) during training. This additionally ensures the planner can robustly overcome upstream TRELLIS imperfections (like geometric noise) and output precise blueprints, as demonstrated in the attached [anonymous link](https://postimg.cc/mzYxqKj8).
> > >
> > > Stage 2 (Mesh Synthesis): While external mobility datasets supplement the continuous kinematics, PhysDB acts as the geometric foundation. The original OmniPart architecture was designed primarily for standard game assets and often produces structural artifacts when tasked with complex, physics-aware topologies. Training on PhysDB provides massive geometric and texture regularization. As shown in the newly attached [anonymous link](https://postimg.cc/LJXjb2cp), our model trained on PhysDB suppresses these artifacts and generates superior, high-fidelity meshes for physics-aware objects.
> > >
> > > **Table 4 Metrics and Post-Processing.**
> > >
> > > All metrics in Table 4 were measured on the raw, unadjusted model outputs before any post-processing. The automated verification step (adjusting kinematic limits) is merely a downstream safety fallback used exclusively during actual physics simulations (as shown in our videos). Because both our method and all baselines are evaluated entirely on raw outputs, the comparison is rigorously fair and reflects true generation capabilities.
> > >
> > > **Appendix A Asymmetric Matching Fairness.**
> > >
> > > The difference in the matching step arises solely from the models' distinct output formats:
> > >
> > > - PhysXGen (Baseline): Outputs dense voxel-level properties, which are evaluated directly on voxels.
> > > - PhysForge (Ours): Outputs structured part-level properties, which first requires mathematical part-to-part matching with the ground truth.
> > >
> > > Crucially, once this initial matching is established, the final numerical metrics for both methods are calculated at the exact same voxel level, ensuring consistent and fair spatial evaluation across all baselines.
> > >
> > > We hope these clarifications fully resolve your remaining concerns regarding the soundness and fairness of our framework, and we will integrate these details into the revised manuscript.

---

### Official Review · Reviewer_Dv6a · 2026-03-10

**Soundness:** 3
**Presentation:** 3
**Significance:** 3
**Originality:** 2
**Overall Recommendation:** 3
**Confidence:** 3

**Summary:**

This paper presents PhysForge, a novel framework for generating physics-aware, interactive 3D assets from a single input image. Instead of reconstructing merely a visual shell, PhysForge aims to jointly infer both the geometric structure and the physical properties required for interaction in dynamic environments.
To support this task, the paper also introduces PhysDB, a large-scale open-source dataset containing 150K 3D assets annotated with four-level physical properties.

**Compliance With Llm Reviewing Policy:**

Affirmed.

**Key Questions For Authors:**

What exactly are the scope of manual review and the quality assurance mechanisms in physDB? Out of the 150,000 assets, what proportion was manually reviewed or corrected?

In Table 3, PhysForge (w/o mask) still shows a clear performance gap in both Voxel IoU (47.66) and Bbox IoU (36.32) compared with the full model that uses masks (53.74 and 42.95). Users often cannot provide precise 2D masks. How should this performance gap be interpreted in practice?

**Limitations:**

yes

**Strengths And Weaknesses:**

Strengths：

A reasonable and straightforward two-stage framework design. The paper presents a clear VLM Planning + Diffusion Generation architecture.

The KVI mechanism encodes kinematic parameters into a special voxel representation and jointly models them with geometric latents in the diffusion transformer, enabling joint modeling of geometry generation, and joint parameter generation for articulated parts / joints.

The paper conducts thorough experimental validation at multiple levels. The model is not only compared quantitatively with baselines such as TRELLIS, OmniPart, and Articulate Anything on both planning and generation tasks, but is also evaluated through ablation studies that demonstrate the effectiveness of incorporating physical properties and the KineVoxel mechanism. In addition, the authors showcase the usability of the generated assets in downstream applications such as the RoboTwin simulator and Unreal Engine.

Weaknesses:

Over-reliance on precomputed 3D voxels: In the first-stage planning process, the VLM requires not only 2D images but also 3D voxels generated in advance by external models such as TRELLIS. If TRELLIS produces 3D voxels with severe geometric omissions at the outset, the paper does not sufficiently investigate whether the VLM can still generate a reasonable physical blueprint.

Data quality and noise in PhysDB: The 150,000 annotations in PhysDB are primarily generated automatically by multimodal large models, with only human-in-the-loop spot checking. At such a large scale, it is difficult to guarantee the absolute accuracy of physical properties. The paper lacks a detailed quantitative analysis of the potential annotation noise rate in this dataset.

---

> ### Author Rebuttal · Authors · 2026-03-31
>
> Thank you very much for taking the time to review and for your support. We try our best to address your questions as follows.
>
> **Reliance on precomputed 3D voxels.**  It is true that generating the physical blueprint in the first stage relies on reasonable 3D voxels, and the subsequent geometry generation in the second stage also depends on this initial voxel quality. However, we view this not as a design flaw, but rather an alignment with the prevailing trend in the field. State-of-the-art 3D generative models increasingly adopt two-stage pipelines with voxel representations, as evidenced by a series of recent works such as SparseFlex (ICCV 2025), LATTICE (CVPR 2026), and Sparc3D. Among current open-source models, TRELLIS achieves state-of-the-art performance in voxel generation, which motivated our choice. Furthermore, thanks to the decoupled nature of our framework, our model can seamlessly adapt to even more advanced voxel generation backbones in the future to achieve increasingly stable and robust results against severe geometric omissions.
>
> **Details of the annotation pipeline.** To ensure the reliability of PhysDB, we designed a multi-stage, human-in-the-loop annotation pipeline based on the Gemini model. Initially, we curated a raw pool of 300,000 3D objects from datasets like Objaverse-XL that match our target category distribution (Furniture & Household, Industrial Supplies, Vehicles, Personal Items, Weapons, Tech & Electronics, and Cultural Artifacts). The VLM is provided with holistic and per-part renderings to determine if the object possesses clear physical semantics suitable for standardized annotation. During attribute validation, we implemented hard constraints: for RealWorldSize, we prioritize extracting exact scale values from the original metadata; if missing, the VLM's prediction is strictly verified against a predefined category-specific prior (e.g., a mug is constrained to 0.05m-0.2m), and any out-of-bound predictions are automatically routed to a manual review list. For functional properties, we guided the VLM to a predefined vocabulary of Atomic Affordances inspired by OAKINK2 to prevent hallucination. Regarding manual intervention and data cleaning, we conducted a pilot annotation on 100,000 objects and performed a mandatory 20% manual audit on the subset deemed valid by the VLM. This manual audit yielded a pass rate of approximately 80%, revealing that the rejected instances were primarily affected by VLM reasoning errors caused by models lacking surface textures or containing microscopic parts. Based on these identified failure modes, we formulated explicit rules to pre-filter the remaining unannotated data before processing it through the VLM, ultimately yielding our high-quality dataset of 150,000 assets. Screenshots of our annotation system interface are provided in the [anonymous link](https://postimg.cc/jC9Q8yqb), and the complete automated data processing and cleaning pipeline will be open-sourced upon publication.
>
> **2D mask input.**
> - We would like to clarify that the Voxel IoU and Bbox IoU metrics in Table 3 are influenced not only by the intrinsic rationality of the part planning but also by the specific granularity of the GT annotations. Part decomposition is an inherently ambiguous task, as a single object can be validly segmented in multiple ways, and the GT merely represents one such specific configuration. When a 2D mask is provided, the model is explicitly guided to match the exact granularity of the GT, which naturally results in higher quantitative metrics. Importantly, the ablation comparing PhysForge-bbox (w/o mask) and PhysForge (w/o mask) clearly demonstrates that incorporating physical semantics enables our model to achieve significantly more reasonable and coherent bounding box planning, even in the complete absence of mask guidance.
> - In practical applications, this performance gap does not hinder usability because acquiring precise 2D masks is no longer a bottleneck. Users can easily leverage off-the-shelf interactive vision models, such as the SAM or Nano Banana 2, to effortlessly obtain controllable 2D masks. Consequently, the 2D mask simply serves as an intuitive and highly accessible control signal, empowering users to dictate the exact granularity of the final 3D generation according to their specific interaction needs.

---

> > ### Author Rebuttal · Reviewer_Dv6a · 2026-04-04
> >
> > The authors provided more concrete clarification on the PhysDB annotation pipeline, the scope of manual auditing, and the practical interpretation of the performance gap between the mask and no-mask settings. These responses are helpful.
> > However, I still have a core concern about the robustness of the method when the initial voxel input contains substantial geometric omissions. The current rebuttal mainly explains the rationale for using TRELLIS and the possibility of replacing it with stronger voxel backbones in the future, but it does not yet provide direct evidence that the Stage-1 planner can still produce a reliable physical blueprint under such failure cases.

---

> > > ### Author Response · Authors · 2026-04-07
> > >
> > > We completely agree that providing direct empirical evidence is necessary to address the concern regarding the planner's robustness. To this end, we have provided an additional visualization in the attached [anonymous link](https://postimg.cc/mzYxqKj8) to explicitly demonstrate how our model handles flawed voxel inputs.
> > >
> > > 1. Characteristics of Voxel Failures.
> > > In practice, while TRELLIS rarely produces massive structural omissions, it frequently outputs suboptimal voxels. The most common failure modes are: (1) Scale mismatch: The voxels fail to fill the normalized space (-0.5 to 0.5) and shrink significantly; (2) Internal noise: Generating excessive, redundant internal voxel structures (as seen in the mug example in the provided link).
> > >
> > > 2. Robustness via Data Augmentation.
> > > If the model is trained without accounting for these errors, flawed voxels will indeed mislead the VLM, resulting in redundant and overlapping bounding box predictions (as shown in Column 3 of the figure). To fundamentally resolve this and prevent the VLM from overfitting to the initial voxels, we implement a robust 3D data augmentation strategy during training. This includes random scaling, morphological erosion and dilation, and random rotations applied directly to the voxel inputs.
> > >
> > > 3. Direct Evidence of Robustness.
> > > By explicitly simulating these failure modes during training, our VLM learns to treat the 3D voxels as a coarse spatial reference rather than an absolute ground truth, forcing it to leverage the 2D image as a compensatory visual prior. As demonstrated in Column 4, with our augmentation strategy, the Stage-1 planner successfully overcomes severe voxel noise and scale errors to output highly precise part-level blueprints, which ultimately guide the generation of accurate downstream meshes and physical properties.
> > >
> > > We hope this direct evidence alleviates your concerns regarding the framework's robustness against upstream voxel failures. We will explicitly include this visualization and discussion on the data augmentation strategy in the supplementary material of the revised manuscript.

---

### Official Review · Reviewer_rCRd · 2026-03-12

**Soundness:** 4
**Presentation:** 4
**Significance:** 3
**Originality:** 3
**Overall Recommendation:** 5
**Confidence:** 3

**Summary:**

This paper introduces a novel architecture for generating physically-grounded 3D assets that leverages a blueprint style approach of generation with a hierarchical setup, and trains on a new dataset called PhysDB.

The PhysDB dataset contains 150,000 assets with physical annotations that compared to other datasets like Objaverse additional contain information such as part level materials/semantics, functional properties, and affordances. The dataset is produced by first passing an object through a multi-modal LLM for an initial annotation, followed by manual human labelling/editing. Assets are primarily sourced from Objaverse, Partnet mobility, and Infinite Mobility

For the model, a VLM is finetuned to predict the more fine-grained annotations of PhysDB by taking as input an image, TRELLIS generated voxels, and PartField generated voxel features.

To generate the actual asset, a diffusion model is trained to generate the final 3D object, based on OmniPart. The paper introduces KVI (KineVoxel Injection) to aid the diffusion process and predict joints to construct articulated objects. Alongside the latents, a KineVoxel is constructed (comprised of joint origins, axes, and motion limits) and processed with the latents in the denoising process after passing the KineVoxel through a 2 layer MLP KineVoxel encoder/decoder.

**Compliance With Llm Reviewing Policy:**

Affirmed.

**Final Justification:**

See my rebuttal acknowledgement for why score was unchanged. Additionally I did not give a 6 because I do not see feel this paper is extremely novel with great impact, in addition to the fact that I give my confidence rating a 3 since I cannot accurately judge a paper outside of my domain of expertise.

**Key Questions For Authors:**

- Are there details about the scaling factors for scaling the KineVoxel? I didn't find any references to them again in the paper, how are they chosen?
- How fast is it to generate a new asset from an image?
- The system only takes a single image as input, however this can easily lead to ambiguities in scale and hide components of some object. Would this system be extendable to multi-view unposed images?

**Limitations:**

No limitations are discussed in the paper. Perhaps some discussion about issues that stem from single image asset generation can be discussed, especially when ambiguities can be problematic.

**Strengths And Weaknesses:**

Strengths:
- An overall strong paper with excellent results compared to baselines. A big improvement is the ability to really generate the parts/articulated objects without relying on sampling asset repositories like some prior work.
- The methodology and figures are straightforward, motivate very clearly why PhysForge outperforms past work by introducing some better features combined with better base architectures/approaches (e.g. OmniPart).
- The new dataset is impressive with such detailed annotations, I hope both this dataset and model can be open-sourced for the community.
- Paper shows good applicability to robotics by importing and playing with generated assets in simulation

Weaknesses:
- The limitation to just using a single image may lead to errors arising due to ambiguities.

---

> ### Author Rebuttal · Authors · 2026-03-31
>
> Thank you very much for taking the time to review and for your support. We try our best to address your questions as follows.
>
> **Single image input.** We designed our framework to rely on a single image primarily to maximize user convenience, as it is highly accessible for users to capture a photograph or use generative AI to create a single input image, whereas obtaining strictly consistent multi-view images in practice is often challenging. Nevertheless, we completely agree that extending the system to support multi-view unposed images is a highly valuable research direction to resolve visual ambiguities and hidden components. Theoretically, our architecture could be adapted to this setting by replacing the single-image condition with multi-image feature inputs. We believe addressing the specific technical challenges of this extension is an important area for future work.
>
> **Scaling factors for KineVoxel.** We thank the reviewer for pointing out this missing detail. To determine the scaling factors, we first analyzed the numerical distribution of the geometry latents, which roughly spans from -8 to +8. To align the KineVoxel with this latent distribution, facilitate the diffusion transformer training, and preserve the precision of the kinematic parameters, we empirically set $S_A$ to 8 and both $S_O$ and $S_L$ to 10. We assigned a slightly larger scaling factor to the origin and limit because they are continuous values uniformly distributed between -1 and 1, whereas the axis typically consists of discrete values like 0 or 1. We will include these specific values and their justification in the implementation details of the revised manuscript.
>
> **Generation speed.** The generation speed depends on the number of parts of the target object. On average, the total inference time to generate a completely new asset from a single input image is approximately 70 seconds, which consists of 25 seconds for the first-stage physical blueprint planning and 45 seconds for the second-stage geometry and kinematic parameter generation.
>
> **Limitation.** We sincerely thank the reviewer for pointing this out. We will explicitly add a Limitations section in our revised manuscript to discuss the inherent challenges associated with single image asset generation. Extending our framework to support multi-view inputs to effectively resolve these ambiguities is indeed a highly valuable direction for future work.

---

> > ### Author Rebuttal · Reviewer_rCRd · 2026-04-05
> >
> > All concerns addressed

---

> > > ### Author Response · Authors · 2026-04-07
> > >
> > > We sincerely thank you for your positive evaluation and for confirming that our rebuttal has fully resolved your concerns.

---

### Official Review · Reviewer_PiE5 · 2026-03-13

**Soundness:** 2
**Presentation:** 3
**Significance:** 2
**Originality:** 3
**Overall Recommendation:** 4
**Confidence:** 4

**Summary:**

This paper studies physics-grounded 3D asset generation from a single image. It introduces PhysForge, a two-stage framework in which a vision-language model first predicts a hierarchical physical blueprint, followed by a Flow Transformer that generates geometry and articulation parameters conditioned on a kinematic representation. The paper also presents PhysDB, a dataset of 150k objects with multi-level physical annotations, including object-level scale and scene information, as well as part-level static, functional, and interactive properties. Experimental results show improvements over several baseline methods.

**Compliance With Llm Reviewing Policy:**

Affirmed.

**Final Justification:**

After considering the rebuttal, which clarifies the dataset construction and baseline comparisons, I am inclined to revise my score from weak reject to weak accept.

**Key Questions For Authors:**

1. The paper briefly describes the dataset annotation pipeline. Could the authors clarify which multimodal model(s) were used during the PhysDB annotation process and whether multiple models or stages were involved?
2. How were physical attributes such as RealWorldSize or functional properties validated during dataset construction? Were they inferred purely from automated models, or were additional constraints or verification procedures applied?
3. Could the authors clarify the role of manual inspection or filtering in the dataset creation pipeline? In particular, what types of errors were most commonly encountered during automatic annotation?

**Limitations:**

The limitations of the proposed dataset and framework are not explicitly discussed. It would be helpful for the authors to include discussion of potential limitations, such as the reliance on automated multimodal annotation pipelines which may introduce noise or bias, the focus on object- and part-level static attributes which may miss complex dynamic behaviors, the dependence on single-image predictions which may fail when articulation cues are ambiguous, and the unclear generalization to out-of-distribution objects or more complex multi-object systems.

**Strengths And Weaknesses:**

**Strengths**

1. The decomposition into physical planning and kinematic realization is reasonable. The use of a VLM for semantic planning and a diffusion model for precise continuous parameter generation is intuitive and technically sensible.

2. Across experiments, the method shows clear improvements over prior baselines.


**Weaknesses**
1. While the proposed PhysDB dataset is a central component of the paper, its relationship to existing datasets is not clearly articulated. In particular, it would benefit from a clearer positioning relative to prior physics-grounded datasets such as PhysXNet, as well as large-scale 3D asset collections like Objaverse, Objaverse-XL, and TexVerse.
2. Several details of the annotation pipeline remain unclear, including the specific multimodal model used for annotation, the criteria used to validate attributes such as RealWorldSize, and the extent of manual filtering or correction. These omissions make it difficult to assess the reliability and reproducibility of the dataset.
3. The experimental section evaluates the method against several baselines, but does not discuss or position the approach relative to some recent works that address related problems in physics-aware or articulated 3D generation, such as PhysX-3D (NeurIPS 2025) and GaussianArt (3DV 2026).
4. In Figure 2, Part-2 appears to correspond to the faucet body rather than the function "control water flow," which may cause minor confusion regarding the part-function mapping.

---

> ### Author Rebuttal · Authors · 2026-03-31
>
> Thank you very much for taking the time to review and for your support. We try our best to address your questions as follows.
>
> **Relationship between PhysDB and other existing datasets.** The raw 3D assets in PhysDB are primarily sourced from Objaverse-XL and other online 3D model repositories, from which we selected categories suitable for physical property annotation based on their metadata. Specifically, our dataset features a balanced distribution composed of Furniture & Household (40%), followed by Industrial Supplies, Vehicles, Personal Items, Weapons, Tech & Electronics, and Cultural Artifacts which each account for 10% of the total distribution. While large-scale collections like Objaverse, Objaverse-XL, and TexVerse only provide static 3D shapes or simple metadata, PhysDB enriches these assets with comprehensive physical property annotations. Furthermore, compared to existing physics-grounded datasets such as PhysXNet, which primarily relies on PartNet data with limited categories and contains only about 26K instances, PhysDB leverages a much broader range of data sources to scale up to 150,000 assets. In addition, our annotation system provides finer granularity by including detailed information such as specific part-level materials and StateMachine.
>
> **Details of the annotation pipeline.** To ensure the reliability of PhysDB, we designed a multi-stage, human-in-the-loop annotation pipeline powered by Gemini. We initially curated a raw pool of 300,000 3D objects from sources like Objaverse-XL, covering our target categories (Furniture & Household, Industrial Supplies, Vehicles, Personal Items, Weapons, Tech & Electronics, and Cultural Artifacts). The VLM analyzes holistic and per-part renderings to verify if objects possess clear physical semantics for standardized annotation. During attribute validation, we applied strict hard constraints. For RealWorldSize, we prioritize exact metadata scale values; if missing, VLM predictions must fall within category-specific priors (e.g., a mug is 0.05m to 0.2m), routing any out-of-bound cases to manual review. To prevent hallucination in functional properties, we restricted the VLM to a predefined vocabulary of OAKINK2-inspired Atomic Affordances. For quality control, we conducted a pilot annotation on 100,000 objects with a mandatory 20% manual audit on the VLM-approved subset. This audit achieved an 80% pass rate, showing that rejections mainly stemmed from VLM reasoning errors on textureless meshes or objects with microscopic parts. We translated these failure modes into explicit pre-filtering rules for the remaining data, ultimately yielding our final 150,000 high-quality assets. Screenshots of our annotation interface are provided in the [anonymous link](https://postimg.cc/jC9Q8yqb), and the complete automated processing and cleaning pipeline will be open-sourced upon publication.
>
> **More baselines.** The baseline denoted as "PhysXGen" in Table 1 and Table 2 of our manuscript is indeed the generation method proposed in PhysX-3D (NeurIPS 2025), and all related evaluations were conducted using their officially released checkpoint. Regarding GaussianArt (3DV 2026), its problem setting is fundamentally different from ours. GaussianArt is a reconstruction method that requires multi-view observations of an object captured across different articulation states to optimize its 3D Gaussian representation. In contrast, our framework is a generative model designed to operate efficiently on a single-image input. Due to this significant disparity in input requirements and task formulations, a direct quantitative comparison is not feasible. However, we will explicitly discuss GaussianArt and clarify these distinctions in the related work section of our revised manuscript. Furthermore, we have provided additional qualitative visual comparisons with the concurrent generation work, PhysX-Anything, in the attached [anonymous link](https://postimg.cc/k2w9xQK8).
>
> **Figure 2.** In Figure 2, the pointer for "Part-2" is actually directed at the handle of the faucet rather than the main faucet body, meaning the function to "control water flow" is accurate.
>
> **Limitation.** We sincerely thank the reviewer for pointing this out, and we will explicitly include a Limitations section in the revised manuscript. While our constructed PhysDB dataset provides rich physical and functional property information, it currently does not contain precise continuous kinematic parameters. This is primarily due to the inherent difficulty of annotating precise numerical axes at scale, as well as the fact that many real-world examples simply do not possess moving parts. Determining how to efficiently scale up the annotation of accurate kinematic parameters remains a valuable direction for future research. Additionally, extending our framework beyond single-image predictions to support multi-view unposed images is another promising avenue to further resolve potential visual ambiguities.

---

> > ### Author Rebuttal · Reviewer_PiE5 · 2026-04-07
> >
> > The rebuttal addressed my main concerns, and I decide to raise my score to Weak Accept.
> > Regarding Weakness 4, Figure 2 appears to use inconsistent part indices between Stage-1 and Stage-2: in Stage-1, the faucet body is labeled as part-3, whereas in Stage-2 it is labeled as part-2. I suggest the authors align the indexing across both stages in the revised version.

---

> > > ### Author Response · Authors · 2026-04-07
> > >
> > > We sincerely thank you for raising your score and confirming that our rebuttal addressed your main concerns. We deeply appreciate your careful observation regarding Figure 2, and we will ensure the part indices are aligned across both stages in the revised manuscript.

---

### Decision · Program_Chairs · 2026-04-30

**Decision:**

Accept (regular)

**Comment:**

This work proposed a framework for interactive asset generation. A VLM predicts a Hierarchical Physical Blueprint. A diffusion model then generates geometry and kinematic parameters according to the blueprint. A large-scale dataset is introduced to support the training of the proposed method.

After the rebuttal period, three reviewers recommended towards acceptance (Weak Accept, Weak Accept, Accept). One reviewer kept the same weak reject rating after the discussion.

The authors did a good job clarifying reviewers' concerns on the dataset construction, baseline comparisons, evaluation details, and roles of components in the proposed method. The decision is to recommend the paper for acceptance. The authors are encouraged to revise the paper to incorporate the comments from the rebuttal period in the final version.